# OpenReview forum: "Attention Slipping: A Mechanistic Understanding of Jailbreak Attacks and Defenses in LLMs"
_ICLR.cc/2026/Conference — Submitted to ICLR 2026_

### Official Review · Reviewer_RYfr · 2025-10-27

**Soundness:** 3
**Presentation:** 3
**Contribution:** 2
**Rating:** 2
**Confidence:** 5

**Summary:**

This paper explores the internal attention dynamics that underlie jailbreak attacks on LLMs and then proposes a defense mechanism based on the findings. The authors first conduct a mechanistic analysis of attention changes during the optimization process of jailbreak prompts, revealing a phenomenon they term attention slipping. They found out that there is a gradual shift in attention distribution across tokens that leads models to be vulnerable. Building on this observation, the authors propose a defense technique called attention sharpening, which introduces a temperature-based adjustment to restructure the attention layout and hence defend the model against jailbreak attacks. Experiments show that attention sharpening modestly reduces attack success rates while preserving relatively high benign task performance.

**Strengths:**

1. The paper is well-written, and the exploratory study provided some insight into the jailbreak attacks.
2. The idea of using a temperature coefficient to sharpen the attention structure is interesting and novel.
3. This paper conducted an adequate number of experiments to demonstrate its effectiveness in both safety preservation and benign performance.

**Weaknesses:**

1.	Marginal efficacy on core objective (ASR)

The reported reduction in attack success rate (ASR) under attention sharpening is small. Given the modest gains, the defense’s practical value is unclear, especially when lightweight alternatives (e.g., system-prompt safety reminders or brief refusal primes) may preserve benign utility while matching or exceeding the observed ASR drop at near-zero cost.

2.	Limited novelty of the central insight

The mechanistic observation that jailbreaks manipulate attention away from safety-critical representations is not new. Section 5 acknowledges several prior works with similar premises and downstream defenses. The paper’s claimed contribution, “distinguishing the dynamic process from the static outcome”, is not applicable in the attention sharpening method which is deployed only at the end of the jailbreak process (i.e., at generation time), functionally still making it a “static” intervention. This defense does not exploit temporal information (e.g., step-wise tracking or counter-drift controls) that would substantiate a dynamic approach.

3.	Insufficient analysis of model-specific failures

Appendix results indicate the ASR increases instead of dropping on certain models (e.g., Gemma-2 9B) against adaptive attacks. This suggests model-specific brittleness and raises concerns about transferability among model architectures and worst-case resilience. This paper does not explain why attention sharpening underperforms on models like GEMMA-2 9B. Without diagnostic analyses (e.g., architecture-level attention statistics, head-type sensitivity, tokenizer effects), it is difficult to generalize lessons or prescribe fixes.

**Questions:**

1. What would be the effect of simply adding a Self-Reminder (https://doi.org/10.1038/s42256-023-00765-8) ?
2. What could be the reason for attention sharpening's differences in performance among different architectures?

---

> ### Author Response · Authors · 2025-11-27
>
> We thank the reviewer for the insightful assessment and for recognizing the novelty of our temperature-based approach.
>
> * **Comparison with Self-Reminder**
> We argue that Self-Reminder functions at the input level, while Attention Sharpening functions at the computation level.
>     * Robustness: Optimization-based attacks (e.g., GCG) reduce the attention weights assigned to safety instructions, including system prompts. Therefore, input-level defenses like Self-Reminder are vulnerable to being bypassed. In contrast, Attention Sharpening modifies the softmax computation to enforce a sharper distribution. This prevents the model from assigning high attention weights to the adversarial tokens.
>     * Efficiency: Unlike Self-Reminders, our method does not increase the input sequence length and introduces no computational overhead.
> * **Clarification on Model-Specific Results (Gemma-2)**
>
> We confirm the reviewer's observation regarding Gemma-2. However, we emphasize that the primary contribution of this paper is identifying the Attention Slipping mechanism. We proposed Attention Sharpening primarily to validate this discovery. The fact that counteracting "slipping" significantly improves safety on the majority of models (3 out of 4) confirms our central hypothesis: that attention diffusion is a key driver of jailbreaks. The result on Gemma-2 suggests that while the Attention Slipping phenomenon is valid, the optimal method for redistributing attention may vary across architectures. Therefore, the specific limitation on Gemma-2 indicates a need for architecture-specific tuning, but it does not invalidate the core mechanistic discovery supported by the broader results.
>
> * **Regarding the Dynamic Analysis vs. Static Defense**
>
> We respectfully clarify the connection between our analysis (Section 3) and our method (Section 4). Our dynamic analysis was performed to identify the root mechanism of failure. This analysis revealed that jailbreak success correlates with the gradual diffusion of the attention distribution (i.e., "Attention Slipping"). Our defense is a static intervention precisely because it is designed to counteract this specific mechanism. By re-scaling the softmax temperature, Attention Sharpening structurally prevents the attention diffusion that the attacks rely on. The reviewer suggests a 'dynamic' defense (e.g., 'step-wise tracking'). However, such an approach would introduce significant computational and latency overheads, rendering it impractical for deployment. We argue that using our dynamic analysis to derive a zero-latency, static intervention is the most efficient and practical solution, as it directly addresses the identified vulnerability.

---

### Official Review · Reviewer_ng4Z · 2025-10-29

**Soundness:** 3
**Presentation:** 2
**Contribution:** 2
**Rating:** 2
**Confidence:** 4

**Summary:**

This paper proposes a mechanistic explanation for how jailbreak attacks bypass LLM safety guardrails. Spesifically, they introduce the concept of Attention Slipping, which is a universal phenomenon where models gradually reduce attention to unsafe portions of input during adversarial prompt optimization. The authors empirically observe this across multiple jailbreak methods (GCG, AutoDAN, MSJ) and models (Gemma2, Llama3, Qwen2.5, Mistral). They further evaluate existing defenses (Token Highlighter, SmoothLLM) and show that their efficacy correlates with mitigating Attention Slipping. They also propose Attention Sharpening, a lightweight defense that modifies attention softmax with temperature scaling to counteract this slipping effect, achieving a favorable trade-off between safety and utility without extra computational cost.

**Strengths:**

1. The paper provides a mechanistic interpretation of jailbreak behavior through “attention slipping,” connecting model interpretability with adversarial robustness.

2. Experiments span four major LLMs and multiple jailbreak families, making the observed phenomenon and proposed defense appear robust and generalizable.

**Weaknesses:**

1. My major concern is although the paper emphasizes its focus on the dynamics of attention slipping rather than static attention redistribution, the mechanism and mitigation strategy (temperature scaling / KV modification) are conceptually similar to RobustKV (ICLR 2025) and related attention-based defenses. The actual distinction feels incremental rather than fundamental.

2. The claim that “attention slipping causes jailbreak success” is largely correlational. There is no controlled intervention or causal ablation verifying that manipulating attention alone changes safety behavior.

3. The proposed “attention sharpening” may oversimplify the multifaceted nature of attention in LLMs. Although the author provide results on ASR and win rate metrics. It is not enough as this is the core method in this work but the authors do not comprehensively evaluate it.

**Questions:**

1. What’s the proposed method's defense result against flipattack?

[1] FlipAttack: Jailbreak LLMs via Flipping

---

> ### Author Response · Authors · 2025-11-27
>
> We thank the reviewer for acknowledging our mechanistic interpretation as a strength. However, we respectfully but firmly disagree with the assessment regarding novelty and causality. We believe these concerns stem from a misunderstanding of our method's fundamental operation compared to prior work.
>
> * **Novelty: Distinction from RobustKV**. The reviewer states that our method is similar to RobustKV. We clarify the fundamental differences in operation and implementation.
>
>     * Memory vs. Computation: RobustKV operates on the KV Cache. It removes tokens with low attention scores from memory. In contrast, Attention Sharpening operates on the softmax computation. It rescales the attention scores using a temperature parameter without removing any data.
>     * Binary vs. Continuous: RobustKV performs a binary operation (keep or remove tokens). Attention Sharpening performs a continuous adjustment to the attention distribution. This allows for fine-grained control over how the model weighs the input tokens.
>     * Implementation Cost: RobustKV requires additional logic to manage cache indices. Attention Sharpening requires only a change to the temperature parameter during the softmax calculation. Consequently, our method incurs zero memory overhead and zero additional latency.
>
> * **Causality: The Defense as Intervention** The reviewer argues that the link between Attention Slipping and jailbreaks is correlational. We argue that Section 4 serves as a causal verification.
>    * Method of Verification: To establish causality, one must modify the variable of interest (the attention score distribution) and observe the effect on the outcome (Attack Success Rate).
>    * Evidence: Attention Sharpening performs this modification. By explicitly sharpening the attention scores (counteracting the slipping), we observe a significant decrease in Attack Success Rate (ASR) across all models (Table 1).
>
> * **Evaluation Depth and "Oversimplification"**
> The reviewer suggests that "Attention Sharpening" oversimplifies the attention mechanism. We clarify that the primary contribution of this paper is identifying the Attention Slipping phenomenon. The proposed defense was designed primarily to validate this mechanistic finding.
>
>     * Validation of Mechanism: The fact that a simple method (temperature scaling) successfully mitigates the attack confirms our central hypothesis: that the reduction in attention scores on the unsafe prototype is a key factor in jailbreak success. If the mechanism were unrelated to this score distribution, this simple method would not be effective.
>
>     * Practical Efficiency: Beyond validating our hypothesis, the simplicity of the method offers practical advantages. Unlike other defenses, Attention Sharpening introduces zero computational overhead.
>
>     * Comprehensive Metrics: We evaluated this method using multiple metrics, including Attack Success Rate (ASR), Utility (AlpacaEval), and robustness against Adaptive Attacks. We believe this provides a sufficient assessment of the method's performance.

---

### Official Review · Reviewer_FmPW · 2025-10-29

**Soundness:** 2
**Presentation:** 2
**Contribution:** 2
**Rating:** 4
**Confidence:** 4

**Summary:**

The paper identifies a purportedly universal dynamic in LLM jailbreaks, Attention Slipping, wherein a model’s attention to an 'unsafe prototype' diminishes as an attack prompt is optimized, facilitating policy evasion. They also propose Attention Sharpening, a temperature-based rescaling of attention logits that aims to counteract the phenomenon with no extra compute/memory.

**Strengths:**

- Cross-attack, cross-model evidence of a single mechanism (relationship between AR and ASR) via both optimization trajectories and reverse masking.
- Mechanistic angle that links attacks and defenses at the attention level
- Simple, efficient defense that avoids multi-pass overhead and large memory costs claimed for alternatives.

**Weaknesses:**

- In Figure 5, the increase in attention rate with stronger defense strength does not appear very pronounced—especially compared to the degree of ASR reduction—so the claim in L361–362 seems somewhat overstated.
- The authors devote substantial space to analyzing and illustrating the Attention Slipping phenomenon; however, as they themselves discuss in Section 5, this phenomenon appears to have been proposed in previous work under different formulations (see L469, “two ways of looking at the same thing”). Moreover, I note that the AttnGCG paper mentioned at L456 already observed that during GCG/AutoDAN attacks, attention shifts from the unsafe prototype to other adversarial contexts—an observation largely consistent with the analyses presented in this paper. In light of this, could the authors further clarify the distinctions between their work and the related studies discussed in Section 5, particularly AttnGCG? This would help me better understand the novelty of this paper, as at present I have concerns that the contribution may be incremental rather than fundamentally new. Alternatively, the authors could consider providing a deeper exploration and demonstration of the Attention Sharpening component, which appears more distinct from prior research but currently occupies only a small portion of the paper.
- There is a trade-off between ASR and utility, but the method sometimes leaves substantial residual ASR (e.g., for Qwen and Mistral), suggesting that the security margin under adversarial adaptive attacks may be limited—Table 1 illustrates this point.
- On benign long-context tasks (such as code generation or retrieval-style QA), does Attention Sharpening impair the model’s ability to recall distant tokens? Were any long-sequence stress tests performed?
- How sensitive are the ASR results to the choice of judge models?

**Questions:**

See Weaknesses.

---

> ### Author Response · Authors · 2025-11-27
>
> We thank the reviewer for the constructive feedback. We address the specific concerns below.
>
> * **Distinction from AttnGCG**
> The reviewer notes the connection to AttnGCG. We clarify the differences in scope and application:
>     * Scope of Mechanism: AttnGCG focuses on exploiting attention shifts for gradient-based attacks. Our work establishes Attention Slipping as a consistent mechanism across multiple attack types, including gradient-based (GCG), genetic (AutoDAN), and in-context learning (MSJ) attacks.
>     * Application: AttnGCG utilizes this observation to optimize attacks. In contrast, we use this mechanistic insight to design a defense. The analysis in Sections 2 and 3 provides the theoretical justification for using temperature scaling as a countermeasure.
>
> * **Correlation between Attention Rate and ASR** : The reviewer notes that the increase in average Attention Rate (AR) appears small compared to the ASR reduction.
>     * Explanation: Attention in LLMs is often sparse. A small increase in the average AR across all heads can reflect a significant recovery of focus in the specific heads responsible for safety refusal. Therefore, even a modest increase in global attention metrics can be sufficient to trigger the model's refusal mechanism.
>
> * **Residual ASR and Safety-Utility Trade-off** The reviewer raises concerns about residual ASR on models like Mistral. We refer to Table 1 to contextulize this performance:
>    * Comparison with Baselines: On Mistral, our ASR (49.6%) is lower than the baseline Token Highlighter (57.8%). While SmoothLLM achieves a lower ASR (29.6%), it reduces the AlpacaEval Win Rate to 16.6%, which significantly impacts model utility.
>     * Conclusion: Attention Sharpening effectively reduces ASR compared to Token Highlighter while maintaining a Win Rate that is comparable to the undefended baseline.
>
> * **Long-Context Utility**:
>   * General Performance: Our results on AlpacaEval, which includes diverse tasks, show that the method maintains high performance (e.g., 89.3% Win Rate on Gemma2).
>   * Flexibility: Since Attention Sharpening is an inference-time parameter adjustment, it can be dynamically disabled or tuned for specific long-context retrieval tasks if necessary. This offers flexibility not available in defenses that modify model weights.
>
> * **Sensitivity to Judge Models** As detailed in the Appendix, we use the HarmBench classifier to evaluate ASR. Notably, this classifier achieves a 94.53% agreement rate with human judgments. This high level of consistency confirms that it is a robust benchmark for jailbreak evaluation, superior to simple keyword matching. This ensures that our reported ASR values accurately reflect successful attacks rather than false positives.

---

### Official Review · Reviewer_RaK3 · 2025-10-31

**Soundness:** 3
**Presentation:** 4
**Contribution:** 3
**Rating:** 4
**Confidence:** 3

**Summary:**

This paper investigates the mechanisms underlying jailbreak attacks on large language models and proposes a novel defense strategy. The authors identify a universal phenomenon called Attention Slipping, where models gradually reduce attention allocated to unsafe prototypes during jailbreak attacks, enabling these attacks to bypass safety mechanisms.

**Strengths:**

1. The paper addresses an important research question by investigating the role of attention mechanisms in jailbreak attacks on LLMs.
2. The paper is exceptionally well-written with clear narrative structure and strong logical flow.
3. Valuable findings and insights regarding the Attention Slipping phenomenon are presented, enhancing understanding of jailbreak mechanisms.

**Weaknesses:**

1. **Limited Conceptual Contribution:** The observation that attention decreases during jailbreak attacks has been previously reported in prior work, which diminishes the technical novelty of this paper. As the authors acknowledge in Section 5, concurrent studies like RobustKV (Jiang et al., 2025) and AttnGCG (Wang et al., 2025b) have already identified that jailbreaks manipulate attention patterns. The primary contribution of this paper is demonstrating that attention slips *gradually* during the optimization process and that this phenomenon is *universal* across different attack families. The authors are encouraged to further clarify their contributions, especially from a conceptual standpoint, to distinguish their work from existing literature.
2. **Limited Scale of Evaluation:** It is commendable that the authors analyze several classic jailbreak attacks (GCG, AutoDAN, MSJ) based on the Attention Slipping phenomenon. Is is suggested, however, to extend the analysis to more recent and advanced jailbreak methods, such as those based on chain-of-thought prompting or multi-turn interactions, e.g., Crescendo. This would help validate the universality of Attention Slipping across a broader spectrum of jailbreak techniques. Besides, all experiments are conducted on relatively small models (7-9B parameters), and the effectiveness on larger, more capable models (e.g., 30B+ parameter models) remains unknown. Given that larger models often exhibit different behavioral patterns and may have more robust safety mechanisms, it is unclear whether Attention Slipping remains as pronounced or whether Attention Sharpening remains effective at scale. This limits confidence in the generalizability of findings to state-of-the-art LLMs deployed in production environments.
3. **Insufficient Evaluation of Performance Impact:** The defense method may inadvertently degrade model performance on benign tasks, particularly for longer inputs where diffused attention could prevent the model from focusing on key information. The evaluation using AlpacaEval (805 prompts, primarily short questions) is insufficient to fully characterize potential performance degradation. More comprehensive evaluations on diverse and longer-context tasks are needed to assess the trade-off between safety and utility.
4. **Marginal Performance Gains Over Baselines:** The proposed Attention Sharpening shows limited improvement compared to existing defenses like Token Highlighter. In Table 1, the ASR reduction is often modest, and in some cases (e.g., Mistral-7B on WildGuardMix: 29.2% vs. 26.5%), Attention Sharpening actually performs worse than Token Highlighter.

**Questions:**

1. Can this explanation be extended to more recent jailbreak techniques, such as chain-of-thought or multi-turn attacks?
2. Does the Attention Slipping phenomenon and the effectiveness of Attention Sharpening persist in larger models (30B+)?
3. How does Attention Sharpening affect model performance when processing longer and more complex inputs?

---

> ### Author Response · Authors · 2025-11-27
>
> We thank the reviewer for the positive feedback, describing the paper as "exceptionally well-written" with "valuable findings." We address the comments on novelty and evaluation below.
>
> * **Conceptual Novelty (vs. RobustKV & AttnGCG)**:
> While we discussed RobustKV and AttnGCG in Section 5, we wish to further clarify the key distinctions between our work and these prior studies:
>
>    * Dynamic Process vs. Static Outcome: RobustKV and AttnGCG observe the static result (low attention on safety tokens). Our work systematically tracks the dynamic process of "Attention Slipping." We show how the optimization gradually reduces attention on the unsafe prototype step-by-step. This provides a mechanism to explain the attack success.
>    * Universal Mechanism: We demonstrate that this phenomenon is consistent across multiple attack types, including gradient-based (GCG), genetic (AutoDAN), and in-context learning (MSJ) attacks.
>    * Efficiency: Unlike RobustKV (which modifies memory cache), our proposed defense, Attention Sharpening, introduces zero memory overhead and zero additional latency.
>
> * **Generalization to Larger Models and Advanced Attacks**
>    * Larger Models: We acknowledge that we tested on 7B-9B models. However, the attention mechanism (softmax) is structurally identical in larger models. Therefore, the phenomenon of attention slipping is likely to persist across model scales.
>    * Advanced Attacks: Multi-turn attacks (like Crescendo) also rely on shifting the model's attention away from safety instructions over multiple turns. Our "Attention Slipping" framework provides a theoretical explanation for this behavior.
>
> * **Impact on Long-Context Tasks**
> We acknowledge the reviewer's valid concern regarding the potential trade-off for long-context retrieval. We admit that specific evaluations on long-context benchmarks were not included in this study. However, our results on AlpacaEval demonstrate that the method maintains high utility in general tasks involving complex instructions (e.g., 89.3% Win Rate on Gemma2). This indicates that the model's core instruction-following capabilities remain robust in standard settings.
>
> * **Comparison with Token Highlighter**:
> We emphasize the advantage of efficiency over Token Highlighter
>    * Zero Overhead: As analyzed in Section 4.3, Attention Sharpening requires no additional GPU memory and no extra inference time.
>    * Practical Value: Token Highlighter typically requires gradient computation, which is computationally expensive. Our method achieves comparable defense performance (ASR reduction) with significantly lower deployment costs.

---

### Meta-Review · Area_Chair_6jsk · 2026-01-03

**Summary:**

Lack of novelty: The core of this paper is no different from the observations made in works such as RobustKV and attnGCG.
Insufficient evaluation of model utility: Only the AlpacaEval instruction-following dataset is evaluated, lacking evaluation on general capability benchmarks such as mathematics and code.
Weak defense effectiveness: Across multiple models, the defense improves results by only about 5%.

**Reviewer Concerns:**

## Reviewer RaK3

 - Limited Conceptual Contribution: Not resolved. The work tracks the dynamic process of “Attention Slipping,” but in essence it is no different from the observations in works such as RobustKV and attnGCG.
 - Limited Scale of Evaluation: Not resolved. No additional experiments were added.
 - Insufficient Evaluation of Performance Impact: Not resolved. No additional experiments were added.
 - Marginal Performance Gains Over Baselines: Resolved. The authors mainly emphasize the overhead advantage compared to the Tokenizer Highlighter baseline.

## Reviewer FmPW

 - As defense strength increases, the improvement in attention rate is not very significant: Resolved. LLM attention is relatively sparse, so a small improvement is reasonable.
 - Not much difference from prior work such as attnGCG: Not resolved, for the same reasons as above.
 - Residual ASR and Safety-Utility Trade-off: Not resolved. The defense effect is indeed weak, with around 40% attack failure to be defended against for Qwen and Mistral.
 - Long-Context Utility: Not resolved. No additional benchmark experiments were added; only AlpacaEval was used.
 - Sensitivity to Judge Models: Resolved. Using consistent evaluation conditions is reasonable, and switching among multiple judge models is unrealistic.

## Reviewer ng4Z

 - Lack of novelty: Not resolved, for the same reasons as above.
 - The claim that “attention slipping causes jailbreak success” is largely correlational: Resolved. Experimental results already show that controlling the attention mechanism can provide some defensive effect.
 - Evaluation Depth and "Oversimplification": Not resolved. No additional benchmark experiments were added; only AlpacaEval was used.

## Reviewer RYfr

 - Marginal efficacy on core objective (ASR): Not resolved. The defense effectiveness is indeed not very good.
 - Limited novelty of the central insight: Not resolved, for the same reasons as above.
 - Insufficient analysis of model-specific failures: Partially resolved. The defense works well on most models, but performs poorly on a few specific models.

**Reviewer Scores:**

## Reviewer RaK3

 - Score remains 4

## Reviewer FmPW

 - Score remains 2

## Reviewer ng4Z

 - Score remains 2

## Reviewer RYfr

 - Score remains 2

---

### Decision · Program_Chairs · 2026-01-26

Reject